# Socioeconomic, demographic and obstetric determinants of maternal near miss in Africa: A systematic review

Mory Diakite[1,2]*, Vincent de Brouwere[2,3,4], Bouchra Assarag[5], Zakaria Belrhiti[2,6], Saad Zbiri[2,6,7], Mohamed Khalis[2,6,8,9]

1 Kankan Regional Hospital, Kankan, Guinea, 2 Mohammed VI International School of Public Health, Mohammed VI University of Sciences and Health, Casablanca, Morocco, 3 Department of Public Health, Institute of Tropical Medicine, Antwerp, Belgium, 4 School of Tropical Medicine and Global Health, University of Nagasaki, Nagasaki, Japan, 5 National School of Public Health, Rabat, Morocco, 6 Mohammed VI Center for Research & Innovation, Rabat, Morocco, 7 Institut d'Analyse des Systèmes de Santé (IA2S), Paris, France, 8 Higher Institute of Nursing Professions and Technical Health, Rabat, Morocco, 9 Laboratory of Biostatistics, Clinical and Epidemiological Research, Department of Public Health, Faculty of Medicine and Pharmacy, Mohamed V University in Rabat, Rabat, Morocco

* mdiakite@um6ss.ma

## Abstract

### Background

High rates of maternal mortality and morbidity remain significant health issues in low- and middle-income countries. Despite this, few researchers have investigated the underlying factors of severe maternal complications in sub-Saharan Africa. Therefore, the objective of this systematic review was to determine the socioeconomic, demographic, and obstetric predictors of severe maternal complications in this region.

### Method

For this systematic review, we searched PubMed, Scopus and Science Direct between 2000 and 2022. Studies were eligible if they reported a relationship between impending maternal death and one or more socioeconomic, demographic or obstetric determinants. We did not contact the authors of the articles, as we had access to their full texts. The quality of qualitative and quantitative studies was assessed using the Critical Appraisal Skills quality assessment tool.

### Results

Among the 3001 identified studies, this systematic review selected 25 articles. Factors such as economic status, level of education, maternal age, marital status, rural residence, transfers to other facilities, and delays during childbirth were identified as the main determinants of severe complications occurrence in Africa. Disparities in access to maternal healthcare were observed among women from different socioeconomic groups, often due to power imbalances in decision-making processes.

**Data Availability Statement:** All relevant data are within the manuscript and its Supporting information files.

**Funding:** The funders had no role in study design, data collection and analysis, decision to publish, or preparation of the manuscript.

**Competing interests:** The authors have declared that no competing interests exist.

## Conclusion

Several factors, including education, prenatal care follow-up, pre-existing medical conditions, method of admission, and mode of delivery, have been identified as significant indicators of the likelihood of severe maternal morbidity. To reduce these cases, it is crucial to implement targeted socio-economic development programs, including improving access to education, strengthening prenatal health services, providing support to pregnant women with pre-existing medical conditions, and ensuring appropriate admission and delivery methods.

## Background

Maternal mortality and morbidity remain a major health problem with a very high ratio in low- and middle-income countries, where 99% of worldwide maternal deaths are recorded, with Sub-Saharan Africa alone accounting for about 66% [1]. Sub-Saharan Africa has the highest maternal mortality ratio with 542 maternal deaths per 100 000 live births compared to 12 deaths per 100000 live births in high-income countries, while the lifetime risk of maternal mortality was 1 in 37, compared to only 1 in 7800 in Australia and New Zealand [2, 3]. The high level of maternal mortality in some parts of the world reflects inequalities in access to health services and highlights the gap between rich and poor, and between rural and urban populations [4].

Moreover, deaths are only the tip of the iceberg, maternal morbidity also represents an enormous burden of disease for women and their families. Research focusing on maternal morbidity, especially near-miss morbidity contributes to better understand maternal death causes and circumstances [5]. A maternal near miss is defined as "a very ill woman who would have died had it not been that luck or good care was on her side" [6, 7]. Women who survived after severe obstetric complications have many similar socioeconomic, demographic, and obstetric characteristics with women who died from these complications [8]. This makes the monitoring and the documentation of maternal near miss very relevant for assessing the quality of maternity services [7–9].

Even when moving to higher levels of care, women with life threatening conditions often experience delays in accessing care. A major problem in Africa is the lack of logistics and health infrastructure. Weak care facilities and lack of appropriate referral networks are also barriers to care. A functional and accessible health system is needed for the prevention and management of maternal near miss (MNM) [10, 11]. Socioeconomic factors remain important factors that impact healthcare access among women. Even in some countries with free healthcare, significant disparities are still shown in the use of healthcare services particularly among mothers and their children [12, 13].

The determinants that may prevent women to benefit from care during pregnancy and childbirth may include characteristics related either to the woman herself, her household, her environment, or her hospital of delivery [4]. Therefore, the identification of determinants during pregnancy and childbirth may be useful to prevent women from dying [14].

### Objectives

The aim of this systematic review was to identify socio-economic, demographic and obstetric factors predictive of maternal near-miss (MNM)in women aged 15–49 years in Africa.

## Method

### Study design

This systematic review was conducted according to the Preferred Reporting Items for Systematic Reviews and Meta-Analyses (PRISMA) [15].

Our review question was to identify what socioeconomic, demographic, and obstetric determinants influence the occurrence of maternal near miss in women aged 15 to 49 in Africa.

### Eligibility criteria

Articles included in this systematic review had to meet the following criteria: they had to present the results of research studies carried out on women who had experienced maternal near-miss in Africa, examine the socio-economic, demographic and obstetric factors influencing the occurrence of these accidents, and be published between January 2000 and December 15, 2022. There were no language restrictions, and no prior restrictions on study design (qualitative, quantitative, or mixed methods). Studies without abstracts or for which the full text was not available were excluded. No studies were excluded on the basis of quality assessment.

This review used the PECO (Population, Exposure, Comparison and Outcomes) approach to research, with the following parameters:

Population: Pregnant, labouring, or postpartum women up to 42 days after delivery in Africa.

Exposure: factors predictive of MNM, such as factors related to the prenatal period (presence of a prenatal visit and frequency of prenatal visit), sociodemographic characteristics and delay in access to care.

Comparison: reference group reported for each predictive factor in each respective study.

Outcome: occurrence of MNM in women.

The adoption of the PECO approach in this review is substantiated by its adeptness in distinctly delineating the Population, Exposure, Comparison, and Outcome parameters, all of which are essential for investigating the predictive factors of severe maternal morbidity occurrence.

### Information sources and search strategy

We searched the following databases: PubMed, Scopus, and Science Direct. The search was carried out to identify relevant articles up to 15 December 2022, using the following keywords or search terms: " life-threatening maternal morbidity", "severe obstetric", "severe maternal morbidity", "maternal near miss", "severe acute maternal morbidity", "socioeconomic status", "risk factors", "Africa". In addition, we included combinations of keywords using Boolean operators which are (OR, AND), but also truncation. The syntax and search equations used in this study are given in the S1 File.

Here is an example of the search equation we used to identify relevant articles in our study.

*«maternal near miss» OR «severe maternal morbidity» OR «severe acute maternal morbidity» OR «life threatening maternal» AND «Africa».*

### Selection process

Zotero [16] reference management software was used to organize and detect duplicate references. To identify eligible articles, we used the PRISMA flow chart. Two authors, MD and BA,

independently reviewed all titles and abstracts from the search, excluding those that did not seem relevant to the topic. The full texts of all potentially eligible articles were retrieved and screened against the inclusion criteria for inclusion in this review.

## Data collection process

Two authors (MD and BA) independently reviewed all included studies to confirm their eligibility. We extracted data using Microsoft Excel for included studies, the authors (MD and BA) independently extracted information such as study characteristics (title, authors, year of publication, country, study design, participants, and sample size) as well as risk factors) (Table 1). Any differences between the authors were resolved by a referee (VDB or MK).

## Data items

Eligible outcomes were grouped into categories including socioeconomic, demographic and obstetric factors. All these factors were considered acceptable for inclusion. In the event of discrepancies in the selection and extraction process, any issues were resolved by discussion, if necessary, with VBD.

**Maternal near miss.** Maternal near miss is defined as "a woman who nearly died but survived a complication that occurred during pregnancy, childbirth or within 42 days of termination of pregnancy" [17], Severe complications include obstructed labor (uterine rupture, imminent uterine rupture, such as prolonged labor after previous caesarean section and emergency caesarean section), severe anemia (haemoglobin < 6 g/dl) and sepsis (postpartum sepsis, chorioamnionitis and septic abortion) [7, 18].

## Study risk of bias assessment

Qualitative studies were assessed for quality using the Critical Appraisal Skills Program (CASP) assessment tool (available at http://www.casp-uk.net) [19]. Quantitative studies were assessed using the National Institute of Health (NIH) quality assessment tool for observational cohort and cross-sectional studies [20]. Study quality assessment criteria included formulation of the research question, study population, population eligibility criteria, sample size justification, outcome measures, consideration of confounding factors, statistical analyses and ethical considerations. In addition, the same NIH tool was used to evaluate the journals included in the study. Following evaluation of these components, all articles were classified into three categories: Good, Fair or Poor.

## Effect measures

In this study, we used the following measures of effect to present our results: the relative risk and odds ratio reported by the studies, along with their 95% confidence intervals. These measures enabled us to assess the association between the variables studied and the outcomes observed, while providing an estimate of the precision of these associations thanks to the confidence intervals.

## Synthesis methods

A modified conceptual framework, based on the work of Domingues et al. (2016), was used to examine the factors that contribute to "maternal near miss" cases in Africa [21]. This framework incorporates socioeconomic, demographic and obstetric factors, which have different

**Table 1. Main characteristics of the included studies.**

| Authors | Title | Years of publication | Study country | Study type | Sample | Results |
|---------|-------|----------------------|---------------|------------|--------|---------|
| Kasahun AW, Wako WG [8] | Predictors of maternal near miss among women admitted in Gurage zone hospitals, South Ethiopia, 2017: A case control study | 2017 | Ethiopia | Case-control study | 229 women (77 cases and 152 controls) | **Socioeconomic and demographics characteristics** • First delay (delay before seeking health care) adjusted Odds Ratio (aOR) = 2,79; 95% Confidence Interval (CI) [1,42; 5,50]) **Pregnancy characteristics** • Mode of admission (referred by facility) (aOR = 7,47; 95% CI [2,27; 24,51]) **Type delivery** • Previous caesarean delivery (aOR = 7,68; 95% CI [3,11; 18,96]) |
| Storeng TK, Murray FS, Akoum MS, Ouattara F, Filippi V [22] | Beyond body counts: a qualitative study of lives and loss in Burkina Faso after near-miss obstetric complications | 2010 | Burkina Faso | Interviews | 64 women | **Socioeconomic and demographics characteristics** • Single • Low income • First delay (delay before seeking health care) **Search of one or more services for delivery care** • Third delay (delay in receiving care) |
| Briand V, Dumont A, Abrahamowicz M, Sow A, Traore M, Rozenberg P, et al. [23] | Maternal and perinatal outcomes by mode of delivery in Senegal and Mali: a cross-sectional epidemiological survey | 2012 | Senegal and Mali | Cross-sectional study | 78166 women | **Type delivery** • Operative vaginal delivery (aOR = 1,7; 95% CI [1,14; 2,66]) • Caesarean delivery (aOR = 3,3; 95% CI [2,83; 3,86]) |
| Liyew EF, Yalew AW, Afework MF, Essén B [24] | Distant and proximate factors associated with maternal near-miss: a nested case-control study in selected public hospitals of Addis Ababa, Ethiopia | 2018 | Ethiopia | Case-control study | 864 women (216 cases and 648 controls) | **Socioeconomic and demographics characteristics** • Single (aOR = 2.38; 95% CI [1.22 4.65]) • Low income (aOR = 2.19; 95% CI [1.43; 3.34]) • Low education (aOR = 3.28; 95% CI [1.85; 5.84]) • Early marriage (aOR = 1.97; 95% CI [1.21; 3.19]) • Rural residence (aOR = 13.0; 95% CI [7.12 23.80]) **Pregnancy characteristics** • Previous hypertension (aOR = 10.80; 95% CI [5.16; 22.60]) • Previous anaemia (aOR = 5.26; 95% CI [2.89; 9.57]) • No Antenatal Care (ANC) (aOR = 10.8; 95% CI [5.16; 22.6]) • Parity > 5 (aOR = 3.53; 95% CI [1.34; 9.27]) • History of stillbirth (aOR = 3.45; 95% CI [1.79; 6.68]) |

*(Continued)*

**Table 1.** (Continued)

| Authors | Title | Years of publication | Study country | Study type | Sample | Results |
|---|---|---|---|---|---|---|
| Worke MD, Enyew HD, Dagnew MM [25] | Magnitude of maternal near misses and the role of delays in Ethiopia: a hospital based cross-sectional study | 2019 | Ethiopia | Cross-sectional study | 572 women | **Socioeconomic and demographics characteristics** <br> • Husband educational status (aOR = 0,29; 95% CI [0,09; 0,96]) <br> • High income (aOR = 0,35; 95% CI [0,18; 0,70]) <br> **Pregnancy characteristics** <br> • No ANC (aOR = 3,16; 95% CI [1,96; 5,10]) <br> • Delay in reaching the final place of care travelled > 10 km (aOR = 1,99; 95% CI [1,10; 3,61]) <br> • Duration of hospital stay ≥ 7 days (aOR = 2,20; 95% CI [1,33; 3,63]) |
| Mengist B, Desta M, Tura AK, Habtewold TD, Abajobir A [26] | Maternal near miss in Ethiopia: Protective role of antenatal care and disparity in socioeconomic inequities: A systematic review and meta-analysis | 2021 | Ethiopia | A systematic review and meta-analysis | 11 studies (98,268 women) | **Socioeconomic and demographics characteristics** <br> • Single (aOR = 1,69; 95% CI [1,03; 2,78]) <br> • Education (aOR = 2,48; 95% CI [1,58; 3,89]) <br> • Residence rural (aOR = 2,7; 95% CI [1,39; 5,26]) <br> **Pregnancy characteristics** <br> • ANC (aOR = 0,33; 95% CI [0,22; 0,49]) |
| Assarag B, Dujardin B, Delamou A, Meski F-Z, De Brouwere V [27] | Determinants of maternal near-miss in Morocco: too late, too far, too sloppy? | 2015 | Morocco | Case-control study | 299 women (80 cases and 219 controls) | **Socioeconomic and demographics characteristics** <br> • First delay (delay before seeking health care) (aOR = 8.71; 95% CI [3.97; 19.12]) <br> • Low education (aOR = 2.35; 95% CI [1.07; 5.15]) <br> • Second delay (delay in reaching to health facility) (aOR = 4.03; 95% CI [1.75; 9.25]) <br> **Pregnancy characteristics** <br> • No ANC (aOR = 3.97; 95% CI [1.42; 11.09]) <br> • Complications during pregnancy (aOR = 2.81; 95% CI [1.26; 6.29]) |

(*Continued*)

**Table 1.** (Continued)

| Authors | Title | Years of publication | Study country | Study type | Sample | Results |
|---|---|---|---|---|---|---|
| Teshome HN, Ayele ET, Hailemeskel S, Yimer O, Mulu GB, Tadese M [28] | Determinants of maternal near-miss among women admitted to public hospitals in North Shewa Zone, Ethiopia: A case-control study | 2022 | Ethiopia | Case-control study | 264 women (88 cases and 176 controls) | **Socioeconomic and demographics characteristics**<br>• Women with the age range of 25–34 years (aOR = 0.26, 95% CI [0.09–0.75])<br>• Women who attended primary and secondary school (aOR = 4.80, 95% CI [1.78–12.90])<br>• Husbands no formal education (aOR = 5.26, 95% CI [1.46–18.90])<br>**Pregnancy characteristics**<br>• Medical disorder during pregnancy (aOR = 12.06, 95% CI [2.82–51.55])<br>• No ANC (aOR = 2.75, 95% CI [1.13–6.72])<br>• Referred (aOR = 4.73, 95% CI [1.78–12.55])<br>**Type delivery**<br>• Caesarean delivery (aOR = 3.70, 95% CI [1.42–9.60])|
| Tariku M [29] | Magnitude of severe acute maternal morbidity and associated factors related to abortion: A cross-sectional study in Hawassa university comprehensive specialized hospital, Ethiopia, 2019 | 2019 | Ethiopia | Cross-sectional study | 345 women | **Socioeconomic and demographics characteristics**<br>• Currently not married (aOR = 0,38; 95% CI [0,22; 0,66])<br>• Uneducated (aOR = 3,3; 95% CI [2,04; 5,33])<br>• Student (aOR = 0,5; 95% CI [0,31; 0,99])<br>• Daily labour and others (aOR = 0,24; 95% CI [0,08; 0,71])<br>• Delay category ≥12 hours (aOR = 4,12; 95% CI [2,24; 7,58])<br>• Residence rural (aOR = 2,00; 95% CI [1,26; 3,15])<br>**Pregnancy characteristics**<br>• Parity ≥4 (aOR = 2,17; 95% CI [1,27; 3,70])<br>• Start of the induced work (aOR = 0,52; 95% CI [0,32; 0,83])<br>• Complete abortion (aOR = 10,86; 95% CI [3,89; 30,27])<br>• Incomplete abortion (aOR = 2,78; 95% CI [1,43; 5,41])<br>• Suspected molar pregnancy (aOR = 6,04; 95% CI [2,76; 13,19])<br>• Intervention surgical (aOR = 3,56; 95% CI [1,65; 7,64])<br>• Referred (aOR = 0,29; 95% CI [0,18; 0,47])|

*(Continued)*

**Table 1.** (Continued)

| Authors | Title | Years of publication | Study country | Study type | Sample | Results |
|---|---|---|---|---|---|---|
| Nansubuga E, Ayiga N, Moyer CA [30] | Prevalence of maternal near miss and community-based risk factors in Central Uganda | 2016 | Uganda | Cross-sectional | 1557 women | **Socioeconomic and demographics characteristics**<br>• Husbands primary education Odds Ratio (OR) = 1.43, 95% CI [1.03–1.98])<br>**Pregnancy characteristics**<br>• Unwanted pregnancy (OR = 1.38, 95% CI [1.06–1.84])<br>• Pregnancy danger signs present (OR = 1.73, 95% CI [1.21–2.46])<br>• Primiparity (OR = 1.83, 95%CI [1.26–2.64]) |
| Gandhi MN, Welz T, Ronsmans C [31] | Severe acute maternal morbidity in rural South Africa | 2004 | South Africa | A prospective audit | 5728 women | **Socioeconomic and demographics characteristics**<br>• Home delivery<br>• No transport available<br>**Pregnancy characteristics**<br>• No or infrequent antenatal care<br>• Essential supplies out of stock<br>• Staff insufficiently trained<br>• Inadequate staffing levels<br>• Substandard prenatal consultation<br>• Incomplete history taken<br>• Delay in diagnosis<br>• Inadequate intrapartum monitoring<br>• Delay in appropriate treatment or referral<br>• Medical records lost |
| Yemane Y, Tiruneh F [32] | Incidence-proportion of maternal near-misses and associated factors in Southwest Ethiopia: A prospective cross-sectional study | 2020 | Ethiopia | A prospective Cross-Sectional Study | 5530 women | **Socioeconomic and demographics characteristics**<br>• Rural residence (aOR = 2,17; 95% CI [1,34; 3,50])<br>• Far distance of living place from hospital (aOR = 2,27; 95% CI [1,34; 3,86])<br>**Pregnancy characteristics**<br>• Primiparous (aOR = 2,19; 95% CI [1,14; 4,22])<br>• Multiparous (aOR = 2,56; 95% CI [1,36; 4,81])<br>• Long-time of diagnosis (aOR = 1,56; 95% CI [1,04; 2,35])<br>• No ANC (aOR = 1,70; 95% CI [1,13; 2,55])<br>**Type delivery**<br>• Duration of labour >18 hours (aOR = 6,89; 95% CI [4,72; 10,01]) |

(*Continued*)

**Table 1.** (Continued)

| Authors | Title | Years of publication | Study country | Study type | Sample | Results |
|---|---|---|---|---|---|---|
| Lemi K, Temesgen T, Demeke K [33] | Determinants of maternal near miss in Western Ethiopia | 2020 | Ethiopia | Case-control study | 183 women (61 cases and 122 controls) | **Socioeconomic and demographics characteristics**<br>• Second delays (delay in reaching to health facility) (aOR = 12.0; 95% CI [2.55, 56.57])<br>**Pregnancy characteristics**<br>• No ANC (aOR = 6.02; 95% CI [1.55;23.28])<br>• Having 2–4 children (aOR = 4.94; 95% CI [1.46;16.80])<br>• Having more than 5 children (aOR = 3.82; 95% CI [1.23; 11.91])<br>**Type delivery**<br>• Going into labour (aOR = 9.40; 95% CI [2.97; 29.71]) |
| Dessalegn FN, Astawesegn FH, Hankalo NC [34] | Factors associated with maternal near miss among women admitted in West Arsi zone public hospitals, Ethiopia: Unmatched case-control study | 2020 | Ethiopia | Case-control study | 321 women (80 cases and 241 controls) | **Socioeconomic and demographics characteristics**<br>• First delay (delay before seeking health care) (aOR = 5.74; 95% CI [2.93; 11.2])<br>**Pregnancy characteristics**<br>• Pre-existing medical conditions (aOR = 2.04; 95% CI [1.11; 3.78])<br>• No ANC (aOR = 3.71; 95% CI [1.1; 12.76])<br>**Type delivery**<br>• Previous caesarean delivery (aOR = 3.53; 95% CI [1.49; 8.36]) |
| Adeoye IA, Onayade AA, Fatusi AO [35] | Incidence, determinants and perinatal outcomes of near miss maternal morbidity in Ile-Ife Nigeria: a prospective case control study | 2013 | Nigeria | Case control study | 375 women (75 cases and 300 controls) | **Pregnancy characteristics**<br>• Chronic hypertension (aOR = 6.85; 95% CI [1.96; 23.9])<br>• Wrong presentation (aOR = 0.16; 95% CI [0.40; 0.67])<br>• ANC (aOR = 0.19; 95% CI [0.09; 0.37])<br>**Type delivery**<br>• Assisted vaginal delivery (aOR = 2.55; 95% CI [1.34; 4.83])<br>**Socioeconomic and demographics characteristics**<br>• Single (aOR = 3.09; 95% CI [1.05; 6.38])<br>• First delay (delay before seeking health care) (aOR = 2.10; 95% CI [1.04; 4.27])<br>• Knowledge of complications (aOR = 0.47; 95% CI [0.24; 0.94]) |

*(Continued)*

**Table 1.** (Continued)

| Authors | Title | Years of publication | Study country | Study type | Sample | Results |
|---|---|---|---|---|---|---|
| Lori JR, Starke AE [36] | A critical analysis of maternal morbidity and mortality in Liberia, West Africa | 2012 | Liberia | A non-experimental, descriptive design utilising maternal death and near-miss audit surveys was utilised for data collection | 120 near-miss events and 28 maternal mortalities | **Socioeconomic and demographics characteristics**<br>• First delay (delay before seeking health care)<br>• Second delay (delay in reaching to health facility)<br>**Search of one or more services for delivery care**<br>• Third delay (delay in receiving care) |
| Habte A., Wondimu M. [37] | Determinants of maternal near miss among women admitted to maternity wards of tertiary hospitals in Southern Ethiopia, 2020: A hospital-based case-control study | 2021 | Ethiopia | Case-control study | 322 women (81 cases et 241 controls) | **Socioeconomic and demographics characteristics**<br>• Second delay (delay in reaching to health facility) (aOR = 3.21; 95% CI [1.61, 6.39])<br>**Pregnancy characteristics**<br>• Pre-existing medical condition (aOR = 2.79; 95% CI [1.45, 5.37])<br>• Lack of a birth and complication preparation plan (aOR = 3.31; 95% CI [1.50, 7.29])<br>• No ANC (aOR = 3.25; 95% CI [2.21, 7.69])<br>**Type delivery**<br>• Previous caesarean delivery (aOR = 3.53; 95% CI [1.79, 6.98]) |
| Geze Tenaw S, Girma Fage S, Assefa N, Kenay Tura A [38] | Determinants of maternal near-miss in private hospitals in eastern Ethiopia: A nested case-control study | 2021 | Ethiopia | Case control study | 432 women (108 cases and 324 controls) | **Socioeconomic and demographics characteristics**<br>• Age ⩾ 35 years (aOR = 2,94; 95% CI [1,37; 6,24])<br>• Age < 20 years (aOR = 2,94; 95% CI [1,37; 6,24])<br>**Pregnancy characteristics**<br>• History of chronic medical disorders (aOR = 2,18; 95% CI [1,03; 4,59])<br>• Anaemia in index pregnancy (aOR = 4,38; 95% CI [2,43; 7,91])<br>• No ANC (aOR = 3,11; 95% CI [1,43; 6,78])<br>**Type delivery**<br>• Previous caesarean delivery (aOR = 4,33; 95% CI [2,36; 7,94]) |

(*Continued*)

**Table 1.** (Continued)

| Authors | Title | Years of publication | Study country | Study type | Sample | Results |
|---|---|---|---|---|---|---|
| Tolesa D, Abera N, Worku M and Wassihun B [39] | Prevalence and associated factors with maternal near-miss among pregnant women at Hawassa university comprehensive specialized hospital, Sida ma region, Ethiopia | 2020 | Ethiopia | Cross-sectional study | 316 women | **Socioeconomic and demographics characteristics** • Time to reach hospital > 60 minutes (aOR = 4,80; 95% CI [1,34; 16,90]) • Residence rural (aOR = 4,2; 95% CI [1,30; 13,9]) **Pregnancy characteristics** • History of stillbirth (aOR = 10,20; 95% CI [1,40; 71,80]) • Referral case (aOR = 5,50; 95% CI [1,80; 17,40]) |
| Turi E, Fekadu G, Taye B, Kejela G, Desalegn M, Mosisa G, et al [40] | The impact of antenatal care on maternal near-miss events in Ethiopia: A systematic review and meta-analysis | 2020 | Ethiopia | A systematic review and meta-analysis | 9 Articles with 5990 participants | **Pregnancy characteristics** • ANC (aOR = 0,28; 95% CI: [0,19; 0,40]) |
| Kalisa R, Rulisa S, van Roosmalen J, van den Akker T [41] | Maternal and perinatal outcome after previous caesarean section in rural Rwanda | 2017 | Rwanda | Retrospective cohort | 4131 women | **Type delivery** • Trial of labour after previous caesarean delivery (aOR = 1,4; 95% CI [1,2; 5,4]) |
| Oppong SA, Bakari A, Bell AJ, Bockarie Y, Adu JA, Turpin CA, Obed SA, Adnu RM and Moyer CA [42] | Incidence, causes, and correlates of maternal near-miss morbidity: A multi-centre cross-sectional study | 2019 | Ghana | Case-control study | 742 women (288 cases and 454 controls) | **Pregnancy characteristics** • Fever in 7 days before delivery (aOR = 5,94; 95% CI [3,64; 9,68]) • Infant birthweight (aOR = 0,99; 95% CI [0.998–0.999]) **Type delivery** • Spontaneous onset of labour (aOR = 0,09; 95% CI [0,05; 0,14]) |
| Tura AK, Scherjon S, Stekelenburg J, van Roosmalen J, van den Akker T, Zwart J [43] | Severe hypertensive disorders of pregnancy in Eastern Ethiopia: comparing the original WHO and adapted sub-Saharan African maternal near-miss criteria | 2020 | Ethiopia | Sub-analysis of a prospective study | 562 women | **Pregnancy characteristics** • No ANC (aOR = 3,17; 95% CI [1,03; 9,76]) • Referred (aOR = 3,34; 95% CI [1,20; 9,31]) |
| Heitkamp A, Aronson SL, van den Akker T, Vollmer L, Gebhardt S, van Roosmalen J, et al. [44] | Major obstetric haemorrhage in Metro East, Cape Town, South Africa: a population-based cohort study using the maternal near-miss approach. BMC Pregnancy Childbirth | 2020 | South Africa | Cohort study | 119 | **Type delivery** • Caesarean delivery (aOR = 4,01; 95% CI [1,58;10,14] |
| Heitkamp A, Vollmer (Murray) L, van den Akker T, Gebhardt GS, Sandberg EM, van Roosmalen J, et al. [45] | Great saves or near misses? Severe maternal outcome in Metro East, South Africa: A region-wide population-based case-control study | 2022 | South Africa | Case-control study | **399** | **Pregnancy characteristics** • Previous hypertension (aOR = 2,4; 95% CI [1,1;5,1] • Caesarean delivery (aOR = 8,4; 95% CI [5,8; 12,3] |

roles in the occurrence of this complication. Each factor is classified as intermediate, distant, or proximal depending on its degree of proximity to the result. Intermediate factors (Fig 1) have a more direct influence on the maternal near miss, while distant factors have an impact on the outcome by influencing intermediate factors.

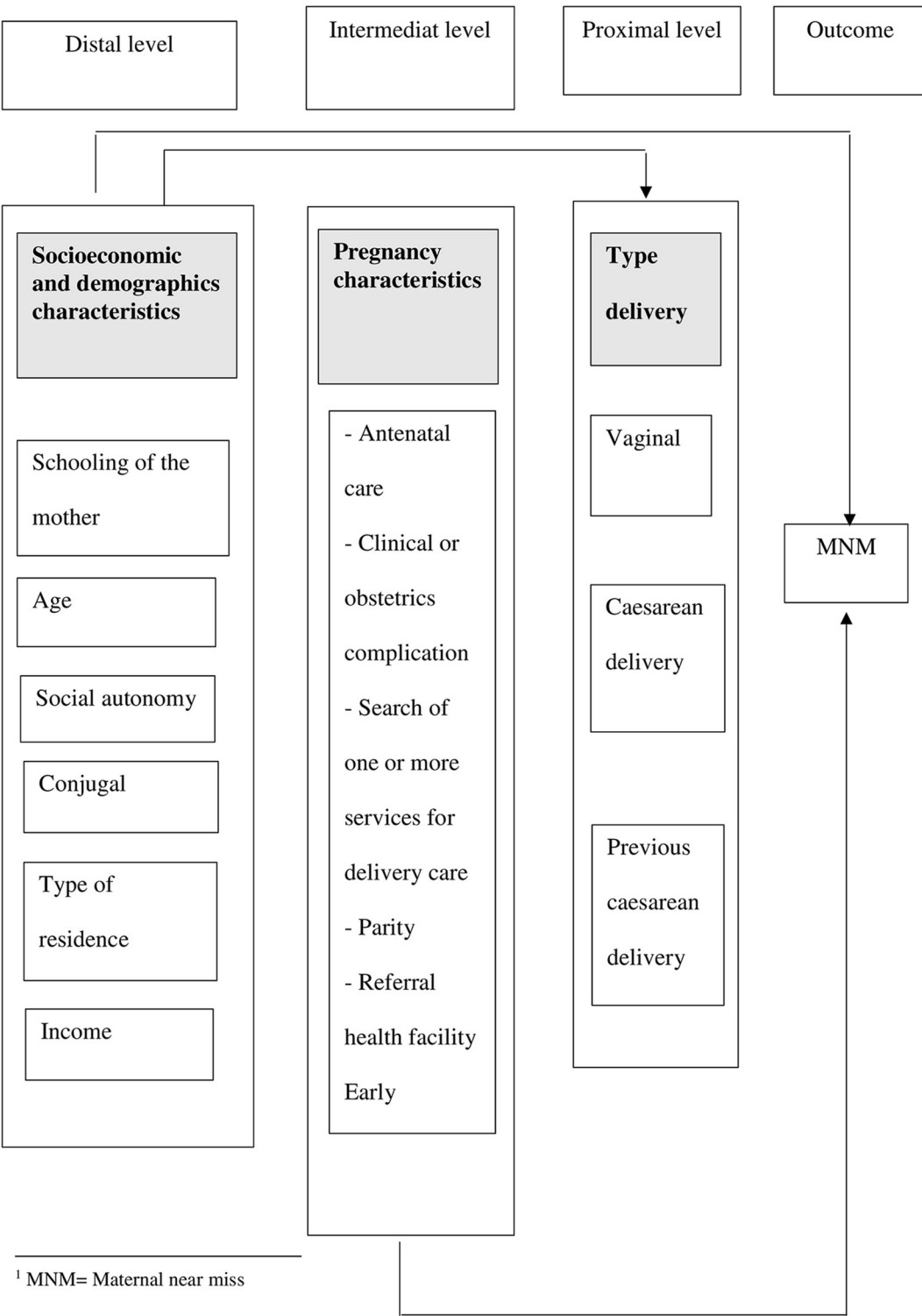

**Fig 1. Framework of the determinants of maternal near miss adapted from Domingues and al.**

## Results

### Study selection

Our initial search identified 3001 articles and 3 additional studies were found by manual search on severe maternal morbidities. We assessed the eligibility of 255 full-text articles after excluding 932 duplicates and 1814 articles not relevant with our research, notably the lack of analysis of the association between the determinants and maternal near miss and studies outside the African continent. We had 230 articles were excluded in the final stage of eligibility assessment, and the remaining 25 articles were included in this study. The reasons for the exclusion of most studies were: irrelevant results, studies that did not report separate results for maternal mortality and maternal near miss.

The selection process and reasons for study exclusion are described in the PRISMA flowchart (Fig 2).

### Study characteristics

In this systematic review, the selected articles were cross-sectional studies (n = 6), cohort studies (n = 2), case-control studies (n = 11), secondary analyses (n = 1), qualitative studies (n = 1), prospective audit (n = 2) and literature reviews (n = 2).

Table 1 describes the characteristics of included studies in this systematic review. The number of participants ranged from 64 in the study conducted in Burkina Faso by Storeng et al. [22] to 78,166 women, in a multi-country study (conducted in Senegal and Mali by Briand et al. [23]). The oldest study considered in this systematic review was conducted in South Africa in 2004, while the most recent studies were from 2022, in Ethiopia.

### Results of individual sources of evidence

**Risk of bias in studies.**   Most quantitative studies were rated as good quality using NIH study quality assessment tools for observational and cross-sectional cohort studies (see S1 Table), as well as for case-control studies (see S2 Table). All qualitative studies were assessed as high quality based on CASP checklists (see S3 Table).

### Results of individual studies

**Socioeconomic and demographic determinants of maternal near miss.**   Poverty [22, 24, 25], maternal education [24, 26–29, 44], husband education [25, 28, 30, 31], delay before seeking health care (first delay) [8, 22, 27, 29, 32–37, 44], maternal age [28, 38], marital status [22, 24, 26, 29, 44], rural residence [24, 26, 29, 31, 32, 39], and delay in reaching the health facility (second delay) [25, 27, 31–33, 36, 37, 39], were the socioeconomic and demographic determinants most frequently associated with the occurrence of maternal near miss in Africa.

Economic insecurity or low socio-economic status played a crucial role in the occurrence of severe maternal accidents in Africa. Women with disadvantaged economic status were more likely to experience such events than their more affluent counterparts. For example, a study by Liyew et al. found that the majority of women who experienced a -maternal near-miss event had a lower monthly income (aOR = 2.19; 95% CI [1.43; 3.34]) [24].

Studies have shown a correlation between women's level of education and the occurrence of maternal near-miss accidents. For example, Assarag et al. reported a marked disparity in female literacy between near-miss cases and controls. They found that the proportion of illiterate women was significantly higher among near-miss cases than among controls, at 65% versus 22% respectively (p<0.001) [27].

**PRISMA 2020 flow diagram for new systematic reviews which included searches of databases and registers only**

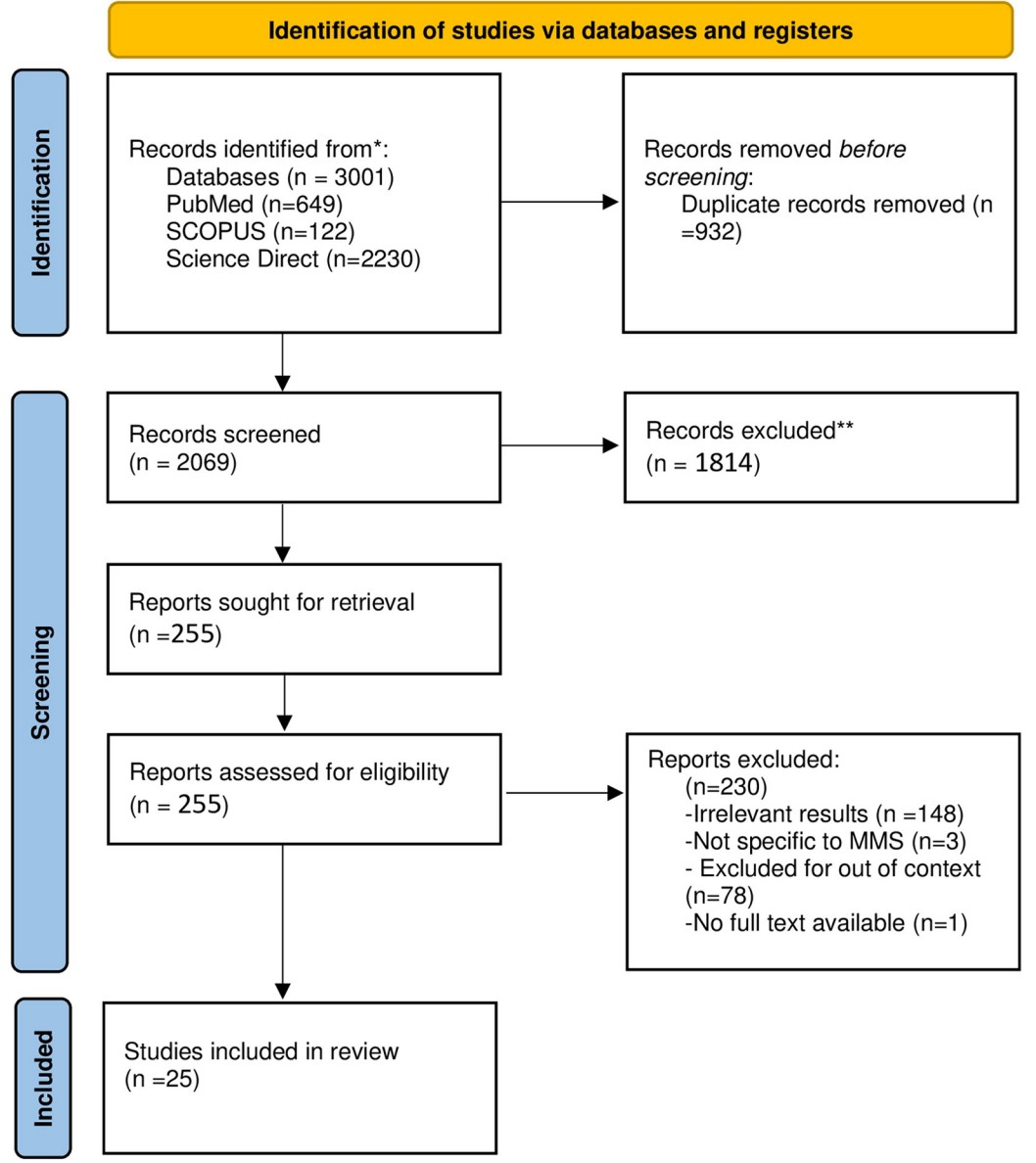

**Fig 2. PRISMA flowchart for the material selection process.**

Lack of financial resources and women's limited autonomy in decision-making have been identified as determining factors in near-miss maternal accidents. For example, Kasahun et al reported that women who waited more than 4 hours for care were nearly 2.8 times more likely to develop a near-miss than those who sought care earlier (aOR = 2.79; 95% CI [1.42; 5.50]) [8].

It has been observed that advanced maternal age (35 years or older) or early maternal age (younger than 20 years) at the time of delivery was consistently associated with an increased

risk of maternal near-miss. A study conducted in Ethiopia found that the likelihood of narrowly averted maternal accident was 2.86 times higher in women under 20 years old and 3.92 times higher in women aged 35 or older [38].

Studies have shown that being without a partner, compared to married women, was independently associated with an increased risk of maternal near-miss. For example, Storeng et al. reported that a significant proportion (a quarter) of the women experiencing maternal near-miss accidents in their sample were unmarried. These women often expressed feelings of guilt after their own families had assumed costs that, in other circumstances, would normally have been borne by their spouses [22].

Several studies have shown that rural residence is strongly associated with an increased risk of missed deliveries. For example, a study by Yayehyirad et al. found that the probability of maternal near-misses was 2.17 times higher in women living in rural areas than in those living in urban areas (aOR = 2.17; 95% CI [1.340–3.500]) [32].

Delay before arrival at a medical center was found to be a determining factor in the occurrence of maternal near misses. Women who experienced a second delay were more likely to suffer a MNM than those who did not. In a study by Lemi et al. it was reported that in 77.05% of cases, it took more than 60 minutes to reach the hospital, while 40.16% of controls also experienced similar delays (aOR = 12, 95% CI: 2.55–56.57) [33]. In addition, Assarag et al. also reported that a higher proportion of women in the near-miss group took more than an hour to reach the first point of care compared with controls (37% vs. 6%, p<0.001) [27].

## Obstetric and healthcare seeking behavior determinants of maternal near miss

Parity [24, 29, 30, 32, 33, 44], lack of prenatal consultation [24–28, 32, 33, 35, 37, 40], and previous caesarean delivery or delivery by caesarean delivery [23, 28, 29, 34, 37, 38, 41, 42, 44, 45], and early maternal age [24] were the main determinants that influenced the occurrence of maternal near miss. Several studies have demonstrated a significant association between caesarean section and the occurrence of maternal near misses. For example, according to an Ethiopian study, women who underwent a caesarean section during their index pregnancy were 7.68 times more likely to suffer maternal morbidity than women who gave birth vaginally (95% CI [3.11 to 18.96]) [8]. Prenatal care plays a significant role in the occurrence of maternal near-misses, as several studies have demonstrated. For example, Lemi et al. showed that mothers who did not receive antenatal care were six times more likely to have major maternal morbidity events (aOR: 6.02 [95% CI: 1.55–23.28]) than mothers who did receive antenatal care [33]. Assarag et al. reported that women who had not received antenatal care were eight times more likely to be near-miss cases (OR = 8.16; 95% CI: 4.08–16.31) [27]. Similarly, a Nigerian study reported that attendance at prenatal consultation in a tertiary facility was protective against maternal near miss OR = 0.19 (95% CI [0.09 to 0.38]), reducing the risk by 5-fold [35].

Studies have identified an association between a history of complications and adverse pregnancy outcomes and the occurrence of near misses [24, 25, 27, 28, 30, 34, 37, 38, 42, 44]. This research has shown that women with pre-existing health problems and a history of adverse events are more likely to experience near-misses. For example, the study by Lyew et al. reported significant differences between cases and controls in the presence of previous medical conditions such as chronic hypertension, anemia and heart problems [24]. A study in Morocco found that women who had experienced complications during pregnancy were 2.81 times more likely to have a maternal near miss than those who had no complications [27]. A study conducted in Ethiopia also observed that women with a history of stillbirth had an increased risk, being 10.2 times more likely than those without a history of stillbirth [39].

Women married at an early age had an increased likelihood of developing severe maternal morbidity, with an odds ratio of 1.97 and a 95% confidence interval of [1.21 to 3.19] [24].

Referral from one facility to another [8, 28, 29, 36, 39, 43, 44] and delay in receiving care (third delay) [22, 25, 31, 36] were the determinants of seeking care delivery services most often associated with the occurrence of a near miss in Africa.

Various studies have highlighted the fact that, among women meeting the criteria for a near-miss, those who were referred had a greater number of serious maternal complications. For example, a study conducted in Ethiopia by Tura et al. found that referred women were 3.3 times more likely to suffer a near-miss than other women (3.34; 95% CI [1.20; 9.31]) [43]. Similarly, a study by Tolesa et al. showed that women referred from another healthcare facility were 5.5 times more likely to suffer a near-maternal event than those not referred (aOR = 5.50; 95% CI [1.80; 17.40]) [39].

## Discussion

This systematic review was conducted to synthesise evidence about determinants influencing the occurrence of severe maternal morbidities in Africa between 2000 and 2022. The information collected focused on the presence of socioeconomic, demographic, and obstetric determinants of maternal near miss. In this systematic review, poverty, maternal education, time spent at home before deciding to go to a health facility, The time elapsed by women between their residence and the healthcare facility, maternal age, marital status, and the place of residence are the most relevant socioeconomic and demographic determinants of maternal near miss. We also identified that obstetric determinants such as parity, lack of prenatal consultation, caesarean delivery and previous caesarean, and pregnancy at young age below 20 years were associated to maternal near miss. Finally, referral from one healthcare facility to another was also associated to maternal near miss, due to the delay in receiving appropriate care.

Extreme poverty and low education levels increase the risk of severe maternal morbidity. Due to various factors, less educated women often have limited access to health-related information. Firstly, they may struggle to read or understand written materials, such as informational brochures or leaflets on maternal health. Secondly, due to financial or geographical barriers, they may be less likely to attend healthcare facilities where health information sessions are organized. Finally, they may face social or cultural norms that restrict their access to education and health-related information. These results are consistent with those of Todd and al. in Afghanistan, who reported that less educated women were more likely to experience a 'maternal near miss' [46]. However, we found an exception: one prospective study conducted in Iran showed that maternal near miss was significantly associated with a higher level of education [47]. The authors of this study explained this association by the fact that women with higher levels of education generally prefer caesarean delivery to vaginal delivery [47]. This is not the case in the African context where women are mainly opposed to caesarean delivery and vaginal delivery is the norm [48]. However, an Iraqi study showed no significant association between women's education and maternal near miss [49]. Women with low income were more likely to experience maternal near miss than richer women. However, findings from included studies conflict with the research conducted in England by Nair et al. [50] who stated that the likelihood of severe maternal morbidity would not vary with regard to socioeconomic status; this may be an indicator of equality in the British National Health System. The correlation between the financial conditions and the cost of high-quality maternal care can explain this association. Financial difficulty is a frequent problem for families, which frequently prevents them from accessing medical facilities or causes delays in access to care, even in case of obstetric

emergency. This is more common in countries where health insurance is essentially private and the coverage very low [13].

The delay in seeking care is a factor that affects the likelihood of maternal near miss. This outcome may be explained by the absence of a family decision-maker, a lack of financial resources, adherence to certain traditional practices such as the consideration of pain and bleeding by some communities as a normal phenomenon, a lack of awareness of danger signs, or a fear of having to pay a significant amount for care [10, 27]. The lack of transport infrastructure forces women to walk long distances, which delays access to medical services and may be responsible for the occurrence of maternal near miss [37].

Women's lack of decision-making ability, adherence to antiquated practices, misunderstanding of danger signs, or worry about having to pay expensive medical bills frequently cause the first delay [10]. Operational issues, such as a shortage of supplies including medicines, blood, equipment, skilled employees, inappropriate attitudes, poorly organized care, or a combination of all these issues, may result in the third delay [10].

Women who did not receive prenatal care were more likely to experience a near-miss event than women who received prenatal care. This result is consistent with studies conducted in Bolivia and Iraq [49, 51]. The most crucial point of contact for mothers who want to speak with medical professionals to learn more about the warning signs during pregnancy and childbirth is prenatal care. If a mother doesn't attend antenatal care consultation, she will not be correctly informed about possible obstetric problems which won't be identified and managed at an early stage [37]. Early and appropriate prenatal care is crucial for controlling pre-existing conditions, keeping track of health, and exchanging health information [52].

According to our systematic review, women who have had a caesarean delivery in the past or who gave birth by caesarean section are more likely to experience maternal near miss [53]. Consistent with previous studies conducted in the Netherlands and northeastern Brazil [54, 55], the risk of maternal near miss was higher among women with a history of cesarean section compared to those without such a history. Although they can save both the mother and the baby's lives, caesarean deliveries increase the risk of infection, bleeding, thromboembolism, uterine scarring, and uterine rupture. Considering the potential risk of caesarean delivery, especially in the African context where access to quality caesarean section is not easy, it is important to maintain the rate within the appropriate range suggested by the WHO (10–15%) [56]. Women with medical issues throughout pregnancy (diabetes, hypertension, cardiac problem) had an increased incidence of maternal near miss. This result is comparable to findings from studies conducted in Iraq and the Netherlands [49, 54]. Additionally, a history of chronic hypertension was also associated with a higher risk of maternal near miss in this study conducted in Brazil [57] In order to lower the frequency of maternal near-miss, pregnant women should be encouraged to check for non-communicable diseases at prenatal consultation appointments [28].

This study shows that referral of pregnant women which is a rational decision in case of need for a higher level of care is associated to maternal near miss. The problem is not the referral itself but the delays. Delays in referral due to lack of transportation or long distance to travel (second delay), inadequate early detection of life-threatening complications can all play a role in near miss of the mother. The reduction of maternal near miss would result from improved reference systems, availability of road infrastructure, and ease of access to maternal health care to the nearest medical facility [28]. The role of reference in the occurrence of maternal near miss cases emphasizes the necessity of strengthening the reference system between first-level and upper-level structures. This could be accomplished through enhancing healthcare providers' ability to communicate with one another, putting in place standard operating procedures,

applying audit standards, enhancing incident reporting, and hiring more personnel to work at the structure level [58].

## Restrictions

One limitation of this review is that most studies came from a single country, Ethiopia, which prevents a more meaningful overall synthesis on the African continent. Studies in multiple contexts can further shed light on the importance of specific determinants, such as organizational, structural, process-related characteristics and provider practices.

## Conclusion

Various factors have been identified as significant indicators of the likelihood of a near miss for the mother, such as level of education, antenatal care, antenatal medical conditions, method of admission and mode of delivery. To reduce the number of near misses, it is crucial to promote socio-economic development. This means implementing measures to improve access to quality education, and to strengthen women's skills and knowledge. Establishing accessible and affordable healthcare systems, strengthening medical infrastructures and promoting health awareness and education. Stimulate employment and encourage income generation, with a particular focus on women.

By taking these steps, we can gradually reduce near-misses and improve maternal health conditions. This requires a comprehensive approach involving coordination and collaboration between governmental players, health organizations, educational institutions and civil society.

## Supporting information

**S1 File. Search strategies.**
(DOCX)

**S1 Fig. PRISMA 2020 flow diagram.**
(DOCX)

**S1 Checklist. PRISMA 2020 checklist.**
(DOCX)

**S1 Table. NIH quality assessment tool for observational cohort and cross-sectional studies.**
(DOCX)

**S2 Table. NIH quality assessment tool for observational and case control studies.**
(DOCX)

**S3 Table. Critical Appraisal Skills Program (CASP) quality-assessment tool for qualitative studies.**
(DOCX)

**S4 Table. NIH quality assessment tool for systematic review.**
(DOCX)

**S5 Table. Complete list of studies identified in the literature search.**
(DOCX)

**S6 Table. Name of data extractors and date of data extraction, as well as confirmation that the study was eligible for inclusion.**
(DOCX)

## Author Contributions

**Conceptualization:** Mory Diakite, Vincent de Brouwere, Mohamed Khalis.

**Investigation:** Mory Diakite, Bouchra Assarag.

**Validation:** Mory Diakite, Vincent de Brouwere, Mohamed Khalis.

**Visualization:** Mory Diakite, Vincent de Brouwere, Mohamed Khalis.

**Writing – original draft:** Mory Diakite, Vincent de Brouwere.

**Writing – review & editing:** Mory Diakite, Vincent de Brouwere, Bouchra Assarag, Zakaria Belrhiti, Saad Zbiri, Mohamed Khalis.

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
