## [Decision Letter · Decision Letter 0]

23 Jan 2024

PONE-D-23-17858Socioeconomic, demographic, and obstetric determinants of maternal near miss in Africa: a systematic reviewPLOS ONE

Dear Dr. DIAKITE,

Thank you for submitting your manuscript to PLOS ONE. After careful consideration, we feel that it has merit but does not fully meet PLOS ONE’s publication criteria as it currently stands. Therefore, we invite you to submit a revised version of the manuscript that addresses the points raised during the review process.

The manuscript has been evaluated by two reviewers, and their comments are available below. Could you please carefully revise the manuscript to address all comments raised?

We look forward to receiving your revised manuscript.

Kind regards,

Avanti Dey, PhD

Staff Editor

PLOS ONE

Journal Requirements:

   "The funders had no role in study design, data collection and analysis, decision to publish, or preparation of the manuscript"

Reviewers' comments:

Reviewer's Responses to Questions

**Comments to the Author**

1. Is the manuscript technically sound, and do the data support the conclusions?

Reviewer #1: Partly

Reviewer #2: Yes

2. Has the statistical analysis been performed appropriately and rigorously? 

Reviewer #1: N/A

Reviewer #2: I Don't Know

3. Have the authors made all data underlying the findings in their manuscript fully available?

Reviewer #1: No

Reviewer #2: Yes

4. Is the manuscript presented in an intelligible fashion and written in standard English?

Reviewer #1: Yes

Reviewer #2: Yes

5. Review Comments to the Author

Reviewer #1: I sincerely appreciate the editorial board of PLUS ONE journal for sending this interesting manuscript to me for review. I have gone through the article, and the followings are my observations and recommendations.

General comments

The review is well written and used appropriate reporting pattern, however I want to draw the attention of the authors to the followings;

Research Topic

1.The research question is well formulated, however the external validity (generalization) may be limited as the words “in Africa” is not well captured by the review as most articles or studies in the review are restricted to few countries or particular region in Africa. The author may need to modify the title or include more studies from other parts of Africa. (I suggest inclusion of nationwide study on near- miss and maternal mortality in Nigeria, https://pubmed.ncbi.nlm.nih.gov/25974281/)

Study Design

2.The Authors expectedly used the PRISMA reporting approach, but no flow chart was included in the review. A PRISMA flow chart that clearly and transparently outline flow of events should be included in the review.

3.The PECO approach was used in designing the research question, the author should include a statement on what informed the choice of this approach as against other approaches such as PICO or SPIDER especially since the review included mixed design methods.

4.Under the Search Strategy and data base search, the authors should include the number of hits or articles from different data bases; PubMed (n=), SCOPUS (n=), Science Direct (n=) etc. Also, inclusion of more data bases (Cochrane, Google Scholar, POPLINE, EMBASE etc) may also increase and capture studies from more countries in Africa which will increase the validity of the review.

5.The quality and bias assessment of the articles included in the study were not explicit enough, such as the criteria used in classifying the articles as being good, fair or poor.

6.The Synthesis of the review is not explicitly clear and Conceptual framework based on Adapted framework by Rosa Maria Soares Madeira Domingue et al used for the synthesis is not included in the review.

Recommendation

7.The consideration of the above issues will improve the quality and robustness of the review.

Reviewer #2: ABSTRACT: Some grammatical errors eg line 30 would better read --- maternal near miss follows similar predictors for ---. The results should be reported in past tense --lines 45-47. The recommendation in lines 51-52 should be more specific.

Line 77, DO the authors mean introduction for the heading of the section instead of rationale?

Lines 90-92 should be referenced.

METHODS: Line 146 should be recasted. In line 183, the authors should specific what they mean by severe bleeding , perhaps also list clinical conditions that cause that. What about other clinical conditions like severe malaria in pregnancy or thromboembolic disorders, where there excluded?

RESULTS: This should be reported in the past tense. eg lines 276-277. Lines 290-293 is written in French and so not understandable. The statement in lines 336-337 appears incomplete and lines 338-340 is not clear.

DISCUSSION-It is not clear in line 369 what the authors mean by - between the residence and birth health care facility ( it does not seem complete). In line 372, the authors should qualify what they mean by young age. The explanation in lines 375-377 is not clear, how women with low educational level will not receive health information . The authors should also include the strengths of their study is there are any.

6. PLOS authors have the option to publish the peer review history of their article (what does this mean?). If published, this will include your full peer review and any attached files.

Reviewer #1: No

Reviewer #2: No

---

## [Author Response · Author response to Decision Letter 0]

21 Mar 2024

Responses to reviewers

Dear Editor,

Please find enclosed the revised version of our manuscript. We would like to thank the reviewers for their constructive comments, which in our view helped us to improve the manuscript.

Please find below our responses to each of the comments.

2. Thank you for providing the following financial information:

"The funders had no role in study design, data collection and analysis, decision to publish, or manuscript preparation."

Question a:

Please specify the sources of funding (financial or material support) for your study. Provide a list of grants or organizations that supported your study, including funding received from your institution.

Answer a:

I hereby confirm that this study has received no financial support from any funder or institution.

Question b:

Indicate the role played by funders in the study. If funders had no role in your study, please state: "The funders had no role in study design, data collection and analysis, decision to publish, or manuscript preparation."

Answer b:

No funders played a role in study design, data collection and analysis, decision to publish, or manuscript preparation.

Question c:

If any authors received salary support from any of your funders, please indicate which authors and which funders.

Answer c:

None of the authors received salary support from any funder for this study.

Question d:

If you did not receive any funding for this study, please state: "The authors received no specific funding for this work."

Answer d:

The authors received no specific funding for this work.

3-When you completed the data availability statement in the submission form, you indicated that you would make your data available upon acceptance. We strongly recommend all authors to decide on a data sharing plan prior to acceptance, as the process can be lengthy and delay publication timelines. Please note that while access restrictions are currently acceptable, all of your data must be freely accessible if your manuscript is accepted for publication. This policy applies to all data unless public deposition would violate the protocol approved by your research ethics committee. If you are unable to comply with our open data policy, please revise your statement to explain your reasoning, and we will seek the editor's advice on an exemption. Rest assured, once you have provided your revised statement, the assessment of your exemption will not delay the peer review process

Answer 3:

As mentioned in my submission form, all data are accessible and have been provided during the process. However, if additional data are needed, please specify the nature of the required data.

4- Please modify the abstract in the online submission form (via Edit Submission) or the abstract in the manuscript so that they are identical.

Answer 4: 

We have adjusted the abstract in the submission form to match exactly the content of the manuscript.

For question 3, "Have the authors made all underlying data supporting the results in their manuscript fully available?" Critic #1: No

We have made all available data accessible. However, if additional data are requested, we would like to know which ones so that we can provide them5. 

1- The research question is well formulated, but the external validity (generalization) may be limited because the phrase "in Africa" is not well captured by the review, as most articles or studies in the review are limited to a few countries or specific regions in Africa. The author may need to modify the title or include other studies from different regions of Africa.

Thank you for this comment. We mentioned Africa in the title, because in this systematic review the search of articles was done for all African countries and we included all articles available and published from any African country. We agree with the reviewer, the generalization of our findings should be interpreted with caution.

To clarify that, we added the following sentence in the limitation section: 

“In this systematic review, most articles are limited to a few countries or specific regions in Africa, the generalization of our findings should be interpreted with caution.”

2- As expected, the authors utilized the PRISMA statement approach, but no flowchart was included in the review. A PRISMA flowchart clearly and transparently describing the sequence of events should be included in the review.

The PRISMA flowchart has been included in this revised version of the manuscript. 

3- The PECO approach was used in designing the research question; the author should include a statement regarding the rationale for choosing this approach over other approaches such as PICO or SPIDER, especially considering that the review involved mixed methods of design.

Thank you for your comment. Our decision to utilize the PECO framework was guided by its alignment with the nature of our exposure-focused study. In this systematic review, we were primarily concerned with the "exposure" of various socioeconomic, demographic, and obstetric factors, rather than specific "interventions". The PECO framework is advantageous in such contexts as it facilitates an examination of how these exposures influence outcomes in our population.

4- The synthesis of the review is not explicitly clear, and the conceptual framework based on the adapted framework by Rosa Maria Soares, Madeira Domingue et al. used for the synthesis is not included in the review.

We added the conceptual framework as supplementary materiel in this revised version of the manuscript. 

NB: We were unable to include this study because the results of severe maternal morbidity and maternal mortality were not differentiated, whereas we were specifically seeking factors associated with severe maternal morbidity (https://pubmed.ncbi.nlm.nih.gov/25974281/).

---

## [Decision Letter · Decision Letter 1]

31 Jul 2024

PONE-D-23-17858R1Socioeconomic, demographic, and obstetric determinants of maternal near miss in Africa: a systematic reviewPLOS ONE

Dear Dr. DIAKITE,

Thank you for submitting your manuscript to PLOS ONE. After careful consideration, we feel that it has merit but does not fully meet PLOS ONE’s publication criteria as it currently stands. Therefore, we invite you to submit a revised version of the manuscript that addresses the points raised during the review process.

We look forward to receiving your revised manuscript.

Kind regards,

Renato Teixeira Souza

Academic Editor

PLOS ONE

Additional Editor Comments:

The manuscript has the potential to make a great contribution to the field; however, it has some limitations that undermine its clarity and reproducibility. Please provide a revised version along with a point-by-point response letter.

Reviewers' comments:

Reviewer's Responses to Questions

**Comments to the Author**

1. If the authors have adequately addressed your comments raised in a previous round of review and you feel that this manuscript is now acceptable for publication, you may indicate that here to bypass the “Comments to the Author” section, enter your conflict of interest statement in the “Confidential to Editor” section, and submit your "Accept" recommendation.

Reviewer #1: All comments have been addressed

Reviewer #3: (No Response)

2. Is the manuscript technically sound, and do the data support the conclusions?

Reviewer #1: Yes

Reviewer #3: Partly

3. Has the statistical analysis been performed appropriately and rigorously? 

Reviewer #1: Yes

Reviewer #3: N/A

4. Have the authors made all data underlying the findings in their manuscript fully available?

Reviewer #1: Yes

Reviewer #3: Yes

5. Is the manuscript presented in an intelligible fashion and written in standard English?

Reviewer #1: Yes

Reviewer #3: Yes

6. Review Comments to the Author

Reviewer #1: (No Response)

Reviewer #3: I understand that this Topic is interesting and that data on LMIC need to be further investigated. The manuscript has improved after revision, but there are still some questions that need to be considered:

-Abstract:

-Methods: only descriptive or a metanalysis was considered? Why wasn´t the review registered?

-Results are not clear and very vague (please include number of cases considered and how the determinants were considered: increased maternal age?? Married or single women were at increased risk?? It is not clear…

-Conclusion has information not previously considered in the results…

-Objectives: Consider using Maternal Near-Miss (MNM) and not “near-maternal acidentes”

-Methods: was the review registered?

-Is reference 17 the best for the definition of MNM, consider the WHO definition-2009 (Pattinson R, Say L, Souza JP, Broek Nv, Rooney C; WHO Working Group on Maternal Mortality and Morbidity Classifications. WHO maternal death and near-miss classifications. Bull World Health Organ. 2009 Oct;87(10):734. doi: 10.2471/blt.09.071001. PMID: 19876533; PMCID: PMC2755324.)

-Flowchart: Please include abbreviations in footnote. Need to be more clear about exclusions: what were irrelevant results? Or out of context?

-Table 1 is interesting, however very descriptive and there is no standard presentation of outcomes… it would be interesting to better describe that and include it as a limitation (variety of outcomes that were not compared- no possible meta-analysis??)

-Discussion should clearly state what variables are specific or different from other continents and what is similar in other settings, especially LMIC.

7. PLOS authors have the option to publish the peer review history of their article (what does this mean?). If published, this will include your full peer review and any attached files.

Reviewer #1: No

Reviewer #3: No

---

## [Author Response · Author response to Decision Letter 1]

27 Aug 2024

Responses to Reviewers:

The following responses are for the Reviewer #3, the only one who made comments to our revised manuscript.

6. Review Comments to the Authors

Question a: “Methods: only descriptive or a metanalysis was considered?”

Answer to a: Not only were descriptive studies and meta-analyses considered, but all studies that examined the factors associated with maternal near miss were also included.

Question b: “Methods: Why wasn´t the review registered?”

Answer to b: Initially, we conducted a scoping review, and at that time, it was not yet possible to register this type of review on the PROSPERO platform. After the first submission of the article, we were advised to transform it into a systematic review and apply the PRISMA guidelines for writing systematic reviews. Unfortunately, it was no longer possible to register it.

Question c: “Is reference 17 the best for the definition of MNM, consider the WHO definition-2009 (Pattinson R, Say L, Souza JP, Broek Nv, Rooney C; WHO Working Group on Maternal Mortality and Morbidity Classifications. WHO maternal death and near-miss classifications. Bull World Health Organ. 2009 Oct;87(10):734. doi: 10.2471/blt.09.071001. PMID: 19876533; PMCID: PMC2755324.) »

Answer to c: We agree that the WHO definition provided by Pattinson et al. 2009 is excellent but its source is “Say L, Souza JP, Pattinson RC. Maternal near miss – towards a standard tool for monitoring quality of maternal health care. Best Pract Res Clin Obstet Gynaecol. 2009;23:287–96. doi: 10.1016/j.bpobgyn.2009.01.007 ». However, we followed your suggestion and cited Pattinson et al. 2009.

Question d: “Flowchart: Please include abbreviations in footnote.”

Answer to d: Done

Question e: “Need to be more clear about exclusions: what were irrelevant results? Or out of context? »

Answer to e: Irrelevant results and out-of-context studies are described on page 11 in the subsection on study selection. 

Question f: “Table 1 is interesting, however very descriptive and there is no standard presentation of outcomes… it would be interesting to better describe that and include it as a limitation (variety of outcomes that were not compared- no possible meta-analysis??)”

Answer to f: We categorized the variables into major groups within each article. This is how the variables were organized. It is true that the lack of a meta-analysis is a limitation of our study.

Question g: “Discussion should clearly state what variables are specific or different from other continents and what is similar in other settings, especially LMIC.”

Answer to g: Done

---

## [Decision Letter · Decision Letter 2]

4 Nov 2024

Socioeconomic, demographic and obstetric determinants of maternal near miss in Africa: a systematic review

PONE-D-23-17858R2

Dear Dr. DIAKITE,

We’re pleased to inform you that your manuscript has been judged scientifically suitable for publication and will be formally accepted for publication once it meets all outstanding technical requirements.

Kind regards,

Renato Teixeira Souza

Academic Editor

PLOS ONE

Additional Editor Comments (optional):

After consulting the editorial office and confirming that a published protocol is not a requirement of the journal in order for this systematic review to be published, I would like to confirm that the article is suitable for publication. I had the opportunity to review it during the last round of revisions, and I recommend it for publication after the authors include minor amendments to address the last minor suggestions.

Reviewers' comments:

Reviewer's Responses to Questions

**Comments to the Author**

1. If the authors have adequately addressed your comments raised in a previous round of review and you feel that this manuscript is now acceptable for publication, you may indicate that here to bypass the “Comments to the Author” section, enter your conflict of interest statement in the “Confidential to Editor” section, and submit your "Accept" recommendation.

Reviewer #3: All comments have been addressed

2. Is the manuscript technically sound, and do the data support the conclusions?

Reviewer #3: Yes

3. Has the statistical analysis been performed appropriately and rigorously? 

Reviewer #3: Yes

4. Have the authors made all data underlying the findings in their manuscript fully available?

Reviewer #3: Yes

5. Is the manuscript presented in an intelligible fashion and written in standard English?

Reviewer #3: Yes

6. Review Comments to the Author

Reviewer #3: Thank you for the responses provided. The manuscript has improved.

-I am still not comfortable with the Paragraph on Line 206, because of the definition of Nearmiss and then “severe complications”, that are not part of the standard maternal morbidity criteria.

-The non- registered review might be a concern for the journal, but thanks for the explanation

-The comment on metanalysis was about the current review. There was no metanalysis performed- just to make it clear that it was a descriptive evaluation of included studies, with the limitation of different outcomes considered…. This was also not reported as a limitation.

-Table 1- 2010 study does not include data on results for socioeconomic characteristics (numbers, analysis… are the characteristics considered associated to MNM?)

7. PLOS authors have the option to publish the peer review history of their article (what does this mean?). If published, this will include your full peer review and any attached files.

Reviewer #3: No

---

## [Editor Report · Acceptance letter]

19 Nov 2024

PONE-D-23-17858R2 

PLOS ONE

Dear Dr. DIAKITE, 

I'm pleased to inform you that your manuscript has been deemed suitable for publication in PLOS ONE. Congratulations! Your manuscript is now being handed over to our production team.

Kind regards, 

on behalf of

Dr. Renato Teixeira Souza 

Academic Editor

PLOS ONE